# Thailand Achievement of SDG Indicator 4.2.1 on Early Child Development: An Analysis of the 2019 Multiple Indicator Cluster Survey

**DOI:** 10.3390/ijerph19137599

**Published:** 2022-06-21

**Authors:** Thitikorn Topothai, Rapeepong Suphanchaimat, Chompoonut Topothai, Viroj Tangcharoensathien, Nisachol Cetthakrikul, Orratai Waleewong

**Affiliations:** 1International Health Policy Program, Ministry of Public Health, Nonthaburi 11000, Thailand; rapeepong@ihpp.thaigov.net (R.S.); chompoonut@ihpp.thaigov.net (C.T.); viroj@ihpp.thaigov.net (V.T.); nisachol@ihpp.thaigov.net (N.C.); orratai@ihpp.thaigov.net (O.W.); 2Division of Physical Activity and Health, Department of Health, Ministry of Public Health, Nonthaburi 11000, Thailand; 3Saw Swee Hock School of Public Health, National University of Singapore, Singapore 117549, Singapore; 4Division of Epidemiology, Department of Disease Control, Ministry of Public Health, Nonthaburi 11000, Thailand; 5Bureau of Health Promotion, Department of Health, Ministry of Public Health, Nonthaburi 11000, Thailand

**Keywords:** child, parents, play, growth and development, cognitive skills, Thailand, MICS

## Abstract

The early years of a child’s life are the foundation for their future capability development. Poor health, hunger, poverty, low parental education, lack of parental interaction, high screen time, and poor housing environment hamper their development. There is little evidence of a link between early child development (ECD) and sociodemographic factors in Thailand. In response to monitoring the achievement of SDG target 4.2.1 (the proportion of young children who are developmentally on track in health, learning and psychosocial well-being) as required by all UN Member States, this study analyses the prevalence of appropriate levels of ECD and its correlates of Thai children aged 3 to 4 years. A cross-sectional study of the 6th Multiple Indicator Cluster Survey (MICS) data in 2019 conducted by the National Statistical Office was employed. Face-to-face interviews with mothers and/or legal guardians were conducted. A total of 5787 children aged 3 to 4 were enrolled in this study. The majority of participants, approximately 92.3%, had achieved an appropriate level of ECD index, defined as children who were developmentally on track in at least three out of these four domains: cognitive, physical, social, and learning. Multivariate logistic regression showed that girls had a higher appropriate development index than boys (Adjusted Odds Ratio [AOR] = 1.56, 95% Confidence Interval [95% CI] 1.28–1.90; children living in the 5th wealth quintile had a higher appropriate index than those in a less well-off family the first wealth quintile (AOR = 2.92, 95% CI: 1.86–4.58. Univariate logistic regression showed children living with parents achieving post-secondary education had a significantly greater appropriate index than children living with parents completing secondary education or below (Crude OR = 1.95, 95% CI 1.47–2.58); children who had appropriate parental interactions of more than four out of six interactions, had a significantly higher chance of having an appropriate index than less than four interactions (Crude OR = 1.52, 95% CI 1.14–2.04). Multi-sectoral policies to support child development in low socio-economic households should be strengthened. In addition, family and community should promote parental interactions through reading and playing with young children. Future studies which directly measure ECD in conjunction with regular monitoring through MICS are recommended.

## 1. Introduction

Early childhood years are critical foundations for a child’s future development. Various early child development (ECD) during their first few years, such as cognitive, motor, language, socio-emotional, and regulatory abilities are fundamental for the fulfilment of their human capability [1,2], and a solid ground for future health and wellbeing [3,4]. This is reaffirmed by the SDG indicator 4.2.1, which is the proportion of young children who are developmentally on track in health, learning and psychosocial well-being [5].

Important factors, namely, health, nutrition, safety, responsive caregiving, and early learning opportunities, are required to ensure that all children grow to become productive citizens [1,2,6,7]. While poor health and malnutrition are key hindrances that limit a child’s growth and development [2,8,9,10,11]. Undernutrition hampers cognitive and motor development, as well as educational attainment [9,10,11]. The negative association between stunting and child development was reaffirmed by Miller et al.’s meta-analysis [9]. Additionally, poverty, low parental education, and a poor-quality home environment are linked to deficits in language and cognitive development [8,12,13,14,15]. Children living in low-socioeconomic families are more likely to have language and cognitive deficits than those living in wealthier families, and these effects can last until their adulthood [2]. Evidence suggests that poverty exposes children to biological and psychological risks, through an alteration of the brain’s structure and function, which can lead to developmental abnormalities and cognitive impairment [13,14,15]. Gender variations in child development can occur as a result of disparities in investment in child health, nutrition, and education [16].

Parents and caregivers can help children attain their full potential by engaging them in a range of activities, including interactions through play [17,18,19]. Children can be positively responded to and interacted with their parents through playing with, talking with, and paying attention to them [20,21]. These interactions provide an opportunity for parents to better understand and communicate with their children in a long run [20,22]. While being exposed to more technology and having access to more electronic devices may cause young children to have disproportionately excessive screen time viewing and hinder their development [22,23]. The World Health Organization (WHO) recommends no time on screen for children under the age of two, and no more than one hour per day for children aged two to four years [24].

Globally, nearly 250 million (43%) children under the age of five in low- and middle-income countries (LMICs) were at risk of not reaching their maximal developmental potential due to stunting or extreme poverty [2,25]; of which 36.8% of 3- and 4-year-old living in 35 LMICs lacked fundamental cognitive and socio-emotional abilities such as following directions and resisting aggression [2]. This evidence sends a strong message for policymakers to monitor and mitigate these challenges through poverty reduction by investing more in nutrition and early childhood education program [26].

In Thailand, a series of ‘Multiple Indicator Cluster Survey (MICS)’ in 2006, 2012, and 2016 [27,28,29] reported approximately 79%, 93% and 93% of children younger than 5 years had achieved appropriate ECD index (calculated as the percentage of children who are developmentally on track in at least three of these four domains: cognitive, physical, social, and learning). Although the national average child development index has been improved, the survey conducted by the Department of Health, Ministry of Public Health in 2007 and 2014 revealed that several children had constantly struggled with delays in language and fine motor development [30]. Besides, there is little evidence, with a large sample size, of a relationship between sociodemographic features, other contributing factors and an appropriate ECD index.

In response to monitoring the SDG target 4.2.1 (the proportion of young children who are developmentally on track in health, learning and psychosocial well-being), this study aims to identify the prevalence of appropriate ECD index and associated factors to this achievement among Thai children aged 3 to 4 years. A better understanding of this relationship can inform policy and call for action in promoting ECD targeting key population groups.

## 2. Materials and Methods

### 2.1. Study Design, Data Source, and Participants

A cross-sectional quantitative design was used in this research. The data obtained in this study was part of the 6th MICS which was jointly conducted by the National Statistical Office (NSO) and the United Nations Children’s Fund (UNICEF) in 2019 [31].

The 2019 MICS sampling method was performed to represent the population at the country level, urban and rural settings, as well as the five regional domains: Bangkok, Central, North, Northeast, and South. The main sampling stratum in each province was classified as urban and rural areas. Within each stratum, a specified number of enumeration areas were systematically selected according to probability proportionate to size. Following the completion of a household list in the designated enumeration areas, systematic random sampling was conducted at the household level. The survey comprised 5787 children aged 3 to 4 years old from these households.

### 2.2. Data Collection, Questionnaire Design, and Variable Management

Face-to-face interviews with mothers and/or legal guardians were performed by the NSO field staff. Each interview lasted about 60 min average approximate. Field employees entered the data into mobile tablets in real-time during the fieldwork. The mothers in the visited households were revisited if they were not physically present during the first round of the survey visit. The data was collected between May and November 2019.

The major dependent variable was the ECD index, calculated as the percentage of children who were developmentally on track in at least three of these four domains: cognitive (literacy-numeracy), physical, social (socio-emotional), and learning. The ECD index was computed by using a 10-item module in the MICS questionnaire [31,32]. Details of the categorization to determine if the children were on track in each domain were presented in Table 1.

Gender, height for age, weight for height, residence location, family wealth, parental education, parental interaction, and screen time, were the main independent variables. Height for age was measured against the standard deviation (S.D.) and was defined into two categories according to the WHO guideline: (a) stunting (<=−2 S.D.) and (b) non-stunting (>−2 S.D.) [33]. Weight for height was also classified into three groups: (i) underweight (<=−2 S.D.), (ii) normal (>−2 S.D. to <2 S.D.), and (iii) overweight or obese (>= 2 S.D.) [33]. There were two types of residential areas: urban and rural. The family wealth index was equally divided into five quintiles from the least well-off to the most well-off. Parental education was classified into two categories: (i) secondary education or less, and (ii) post-secondary education. Six parental interactions in the last three days before the survey date were: (a) book reading, (b) storytelling, (c) singing, (d) identifying or counting or drawing, (e) outdoor play, and (f) family play. We have categorized these parental interactions into two groups; (a) children had parents engaged in at least four out of six interactions as appropriate parental interaction, and (b) children had parents engaged in less than four interactions as inappropriate parental interaction. This is in line with MICS classification and reporting [32,34]. In addition, daily screen duration on electronic device was categorized into two ranges: (i) <=1 h/day, and (ii) >1 h/day according to the WHO recommendation [24].

### 2.3. Data Analysis

There were three stages of data analysis. First, descriptive statistics were employed to provide an overview of the data. Second, a Chi-square test was used to assess the association between each independent variable and childhood development. Third, the independent variables that showed a significance level (*p*-value < 0.05) by the Chi-square test were then further analyzed by the univariate and multivariate logistic regression analysis. A multivariable logistic regression analysis was used to assess the relationship between achieving an appropriate ECD index with all independent variables at the same time. The results from univariate logistic regression will be presented in the form of crude Odds ratios (crude OR) with a 95% confidence interval (CI), while results from multivariate logistic regression will be in the form of adjusted Odds ratio (AOR) with a 95%CI. STATA software version 17 was used for all calculations (serial license number: 401709350741).

## 3. Results

### 3.1. Baseline Characteristics and ECD Index

NSO enrolled a total of 5787 children aged 3–4 years. As demonstrated in Table 2, the majority of participants, approximately 92.3%, were perceived as being on track for development (achieving at least three out of four domains). Girls had a higher proportion of appropriate ECD index than boys (94% versus 91%, *p*-value < 0.001). Children from low-income households exhibited significantly lower levels of appropriate ECD index than those from high-income families (for instance, 88% for the first quintile and 96% for the fifth quintile, *p*-value < 0.001). Children living with parents achieving post-secondary education had greater levels of appropriate ECD index than children living with parents completing secondary education or below (95% versus 91%, *p*-value < 0.001). In addition, children having appropriate parental interactions had significantly higher levels of appropriate ECD index than their counterparts (93% and 89%, respectively, *p*-value < 0.01).

### 3.2. Percentage of Children by Gender and Family Wealth Who Achieved Appropriate ECD Index by Four Domains

A comparison of the achievement of ECD domains by gender was performed. Girls had greater achievements than boys in the overall ECD index (94% versus 91%), cognitive domain (64% versus 61%), and social domains (87% versus 81%)—see Figure 1. While there was a similar ECD index between girls and boys in physical and learning domains.

Children from high-income households exhibited higher levels of overall ECD index than those from low-income families (for instance, 96% for the fifth quintile and 88% for the first quintile); of the cognitive domain (76% for the fifth quintile and 51% for the first quintile); and of the social domain (88% for the fifth quintile and 81% for the first quintile)—see Figure 2. While there was a similar ECD index among children in all income households in physical and learning domains.

### 3.3. Achieving Appropriate ECD Index: Univariate and Multivariate Logistic Regression Analysis

Multivariate logistic regression showed that; (a) girls tended to have a higher level of appropriate ECD index than boys (AOR = 1.56, 95% CI 1.28–1.90; (b) children living in the families belonging to the fifth wealth quintile were more likely to achieve an appropriate ECD index than those in the families belong to the first wealth quintile (AOR = 2.92, 95% CI: 1.86–4.58); (c) children living with parents having post-secondary education had a greater chance to achieve an appropriate ECD index than children living with parents completing secondary education or below (AOR = 1.34, 95% CI 0.98–1.83) though no statistically significant; however it was statistical significance in univariate logistic regression (crude OR = 1.95, 95% CI 1.47–2.58)—see Table 3. In addition, children who had appropriate parental interactions had a significantly higher chance of having an appropriate ECD index in univariate logistic regression (crude OR = 1.52, 95% CI 1.14–2.04), but no statistical significance in multivariate logistic regression (AOR = 1.28, 95% CI 0.95–1.72).

## 4. Discussion

As committed to SDG Target 4.2; by 2030, UN Member States shall ensure that all girls and boys have access to quality ECD, care and pre-primary education so that they are ready for primary education [35]. This study analysed the achievement of SDG indicator 4.2.1 which measures the proportion of young children who are developmentally on track in health, learning and psychosocial well-being in 2019 and identifies barriers to achieving this goal by different households and parental socio-economic profiles.

Approximately 92% of Thailand’s children aged 3–4 years old had an appropriate ECD index, higher than a global average of 75% (range from 36% to 97%) in 80 countries [5]. Thailand was ranked seventh out of 80 countries (girls ranked 7th, while boys ranked 8th) [5]. Compared with the prior prevalence of appropriate ECD in Thailand, 79%, 93% and 93% in 2006, 2012, and 2016 [27,28,29], the prevalence of 92% in this study demonstrated that Thailand can sustain a high level of appropriate ECD index during the last decade though there is still room for improvement, in particular inequity gaps.

In addition, this study identified that cognitive development was the least achieved domain, 61% in boys and 64% in girls (Figure 1). The prevalence of appropriate cognitive domain was positively related to the family wealth (the higher the family wealth, the better the cognitive outcome) (Figure 2). To address inadequate achievement of cognitive domain among the less wealthy families, policy should create awareness for increased parental and caregiver interactions with their young children particularly through reading with them to increase language literacy—a key component of cognitive development [20,21]. All relevant authorities, such as the Ministry of Public Health, Ministry of Education, Ministry of Social Development and Human Security and local governments, should provide funding support for high-quality and a large variety of children’s books in community libraries, ECD centres, community centres, kindergartens and elementary schools. The policy should subsidize the cost of children’s books and make them accessible and affordable to all families regardless of wealth status. This will sustain the achievement of the SDG 4.2.1 indicator while minimizing the rich-poor inequity gaps.

A study in the US shows that a universal monthly child allowance would reduce child poverty by about 40%, deep child poverty by nearly half and would effectively eliminate extreme child poverty [36]. Since 2015, Thailand adopted the Child Support Grant which provides minimum social protection, 600-baht (US $17) monthly cash grant to 2.3 million children under 6 targeting families with a per capita annual income below 100,000 baht. (US $2800). In September 2020, the National Child and Youth Committee approved extending the Child Support Grant to cover every child under 6-year-old (4.2 million children under 6 years old in Thailand) [37]. Despite targeted child grants, 30% of children from poor families eligible for the grant were missed out due to problems with mean testing and registration [38]; a common challenge in reaching the poorest for public assistance [39]. In addition, due to the COVID-19 pandemic which results in sluggish economic growth and limited fiscal space; the Ministry of Social Development and Human Security decided to defer the universal child grant [40].

This study also shed light on contributing factors to achieving an ECD index. Gender, family wealth, parental education, and parental interaction, were four key factors significantly related to children’s developmental status. Differences in investment in child health, nutrition, and parental education can also cause gender differences in child development. Girls had a higher advantage in early communication development while boys had more vulnerable communication system development [41]. In a 2017 study, Weber et al. discovered that girls aged 3–5 years outscored more than boys on development tests in six low- and middle-income nations in the East Asia-Pacific region [16]. Our study is also by a UNICEF assessment of 80 countries’ ECD, which concluded that girls had made more progress than boys (77% versus 73%) [5].

This study discovered that a greater proportion of children from high-income families achieved a higher appropriate ECD index than those raised by the less well-off families. Inadequate family income not only resulted in a lack of financial resources but also created psychological stress for parents to make ends meet and no time for quality interactions with their children [42]. Consequently, children from low-income families potentially faced multiple challenges, such as unhealthy diet and poor nutrition, difficulties in accessing healthcare, and a poor social environment (including violence and drug abuse) [2,8,13,14]. They were also more likely to experience family conflicts, violence, parental separation, instability, and a chaotic home environment [13,14]. A study in Thailand showed higher stunting among children living in the least well-off households [43], suggesting that poverty is a major influential determinant of a child’s nutrition status and subsequent developmental outcome, although height for age and weight for height did not show a significant relationship with appropriate ECD index in this study (Table 2).

Although univariate logistic regression showed statistical significance in having a greater chance to achieve an appropriate ECD index in children living with parents with higher education. This can be explained by parental education’s correlation with others such as wealth and a better social environment in other national surveys in Thailand [44,45,46]. Higher educated parents were more capable of obtaining knowledge, more aware of their children’s sickness and health needs, and had better attention to child-rearing practices and quality interactions with children [47,48]. While lower educated parents may be obliged to work overtime due to financial constraints, leaving them with little opportunity to connect with and promote their children’s development at home, and the poor neighbourhood environment may not be conducive for child development [42,44,49,50,51,52].

This study showed that children who had parental interaction in at least four out of six activities had a significantly higher chance of having optimal cognitive, physical, and overall development (Figure 2). This effect was also demonstrated by univariate logistic regression analysis in Table 3. It could be explained that parental-child interaction provided opportunities for children to learn naturally a variety of skills and enhance their ingenuity (the ability to solve difficult problems), dexterity (hand skill in performing tasks), cognition and physical strengths [22,53]. Several studies highlighted parental interaction through play as a critical component for the development of a healthy brain, and executive functioning skills that are required in the 21st century [17,18,19]. The importance of parental interaction is highlighted by the SDG indicator 4.2.3, which measures the percentage of children under 5 years experiencing positive and stimulating home learning environments [34]. Thus, parents are encouraged to interact more with their children.

This study did not discover statistical significance between using watching time on electronic devices or television and the ECD index. This is in line with ongoing debates in other studies between the benefits and risks of screen time on child development. Some research has shown that electronic devices can help children learn and develop [54,55]. However, constant stimulation and absorption of visual content on screens have also been shown to impact young children’s focus and concentration [56]. Television viewing time has been negatively associated with the development of physical and cognitive abilities, and positively associated with obesity, sleep problems, depression and anxiety in children [57].

This study identified both strengths and limitations. Key strength lies with the large samples from nationally representative households conducted by NSO which provided robust findings to inform progress towards SDG indicator 4.2.1. In addition, a multivariate logistic regression analysis which adjusted multiple factors at the same time diminished bias which may cause by confounding factors.

A few limitations need to be addressed. Firstly, the sampling frame relied on a household registry managed by the Civil Registration Office. Although NSO used this sampling frame in all other national surveys, the registry may not cover the marginalized populations such as illegal migrants, homeless people, and slum dwellers. As a result, the results may slightly overestimate the prevalence of appropriate child development. Secondly, the answer in the survey relied solely on parental understanding and perception of the 10-module of the ECD index without detailed actual developmental observation by the surveyors. The respondent’s report can suffer from social desirability bias which may underreport socially undesirable behaviours and over-report more desirable attributes [58]. An interview survey using a 10-item module of ECD with direct child development observation or measurement is recommended for future studies. Thirdly, certain possible confounders on the ECD index, such as neighbourhood settings, parental attitudes on child-rearing, and familial genetic disorders, were not captured by the questionnaire. Exploring the link between these aspects and the ECD index can broaden the scope of knowledge for policy. Further qualitative research which aims to explain how the ECD index is influenced by many factors, such as parental knowledge, attitude, and barriers towards appropriate child-raising practices, should be conducted and this will complement the regular quantitative surveys through MICS. Ultimately, a clearer understanding of this issue will help optimize policy to promote sustainable and equitable ECD for all Thai children.

We continue to monitor the progress of SDG target 4.2.1 when the database of the next round of MICS to be conducted by NSO is released to the International Health Policy Programme of the Ministry of Public Health. Such continued monitoring will inform policy on the performance of interventions and drive universal child grants.

## 5. Conclusions

This study monitors the progress toward SDG target 4.2.1; as well as sheds light on contributing factors of the appropriate ECD index among Thai children aged 3 to 4 years. Gender, family wealth, parental education, and parental interaction were positively associated with a higher level of appropriate ECD index. Multi-sectoral policies to support child development in lower socio-economic households should be strengthened. In addition, family and community are recommended to promote parental interactions through reading and playing with young children. Future studies which directly measure ECD in conjunction with regular monitoring through MICS are recommended.

## Figures and Tables

**Figure 1 ijerph-19-07599-f001:**
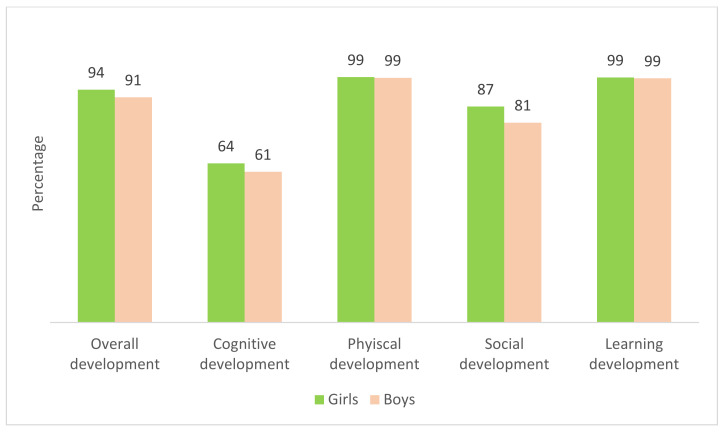
Percentage of children by gender who achieved appropriate ECD index by four domains.

**Figure 2 ijerph-19-07599-f002:**
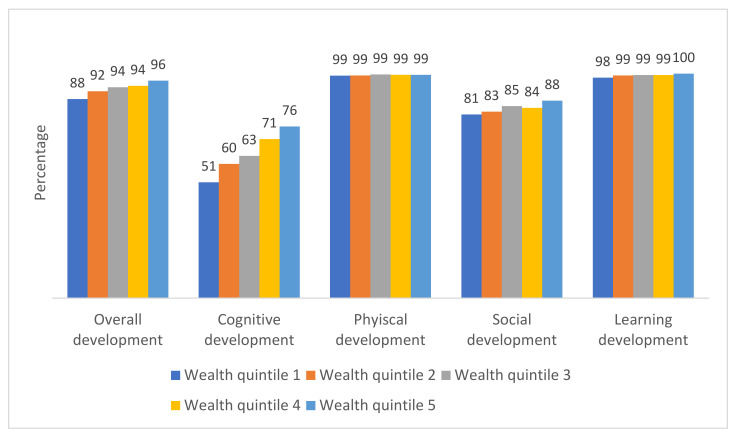
Percentage of children by wealth quintile who achieved appropriate ECD index by four domains.

**Table 1 ijerph-19-07599-t001:** 10-item module for the ECD index in the MICS questionnaire for children aged 3–4 years [5,31,32].

10-Item Module	Four Domains of Developmental Achievement	ECD Index
Can (name) identify or name at least ten alphabets?	Cognitive (literacy-numeracy): children are identified as being developmentally on track based on whether a) they can identify/name at least ten alphabets, b) they can read at least four simple, popular words, and c) they know the name and recognize the symbols of all numbers from 1 to 10. If at least two of these are true, then the child is considered developmentally on track in the cognitive domain.	A child who presents with developmental achievement in at least three of the four domains was then considered “appropriate” for the early child development index.
Can (name) read at least four simple, popular words?		
Does (name) know the name and recognize the symbol of all numbers from 1 to 10?		
Can (name) pick up a small object with two fingers, such as a stick or a rock from the ground	Physical: if the child can pick up a small object with two fingers, such as a stick or rock from the ground and/or the mother/caretaker does not indicate that the child is sometimes too sick to play, then the child is regarded as being developmentally on track in the physical domain.	
Is (name) sometimes too sick to play?		
Does (name) get along well with other children?	Social (social-emotional): children are considered to be developmentally on track if two of the following are true: a) the child gets along well with other children, b) the child does not kick, bite, or hit other children, and c) the child does not get distracted easily. Then the child is considered developmentally on track in this domain.	
Does (name) kick, bite, or hit other children or adults?		
Does (name) get distracted easily?		
Does (name) follow simple directions on how to do something correctly	Learning: if the child follows simple directions on how to do something correctly and/or when given something to do and can do it independently, then the child is considered developmentally on track in this domain.	
When given something to do, is (name) able to do it independently?		

**Table 2 ijerph-19-07599-t002:** Comparing the ECD index by children’s attributes.

Variables	Appropriate ECD Index		Inappropriate ECD Index		*p*-Value
	n	%	n	%	
Total	5342	92.3	445	7.7	
Gender					<0.001
Male	2679	90.8	271	9.2	
Female	2663	93.9	174	6.1	
Height for age					0.30
Stunting	621	91.3	59	8.7	
Non-stunting	4721	92.4	386	7.6	
Weight for height					0.20
Underweight	355	94.2	22	5.8	
Normal	3953	92.0	345	8.0	
Overweight or obesity	1034	93.0	78	7.0	
Residential area					0.17
Urban	1901	93.0	144	7.0	
Rural	3441	92.0	301	8.0	
Family wealth (quintile)					<0.001
1	1238	88.2	165	11.8	
2	1226	91.7	111	8.3	
3	1158	93.5	80	6.5	
4	971	94.1	61	5.9	
5	749	96.4	28	3.6	
Parental education level					<0.001
Secondary education or below	4483	91.4	385	8.6	
Post-secondary education	1304	95.4	60	4.6	
Appropriate parental interaction					<0.01
No	487	89.2	59	10.8	
Yes	4855	92.6	386	7.4	
Screen time					0.74
<=1 h/day	4510	92.4	373	7.6	
>1 h/day	832	92.0	72	8.0	

**Table 3 ijerph-19-07599-t003:** Univariate and multivariate logistic regression analysis of achieving appropriate ECD index.

Variables	Univariate Logistic Regression		Multivariate Logistic Regression	
	Crude Odds Ratio	95% Confidence Interval	Adjusted Odds Ratio	95% Confidence Interval
Gender				
Female	1.55 **	1.27–1.89	1.56 **	1.28–1.90
(ref = male)				
Family wealth				
Quintile 2	1.47 *	1.14–1.90	1.43 *	1.11–1.85
Quintile 3	1.93 **	1.46–2.55	1.84 **	1.39–2.44
Quintile 4	2.12 **	1.56–2.88	1.90 **	1.38–2.61
Quintile 5	3.57 **	2.36–5.38	2.92 **	1.86–4.58
(ref = quintile 1)				
Parental education level				
Post-secondary education	1.95 **	1.47–2.58	1.34	0.98–1.83
(ref = secondary education or below)				
Appropriate parental interaction				
Yes (more than four out of six interactions)	1.52 *	1.14–2.04	1.28	0.95–1.72
(ref = less than four)				

* A *p* value < 0.01, ** A *p* value < 0.001.

## Data Availability

MIC6 surveys. Available online: https://mics.unicef.org/surveys (accessed on 1 January 2022).

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
