# Peer review of "Thailand Achievement of SDG Indicator 4.2.1 on Early Child Development: An Analysis of the 2019 Multiple Indicator Cluster Survey"

_ijerph, 2022, doi:10.3390/ijerph19137599_

Round 1
Reviewer 1 Report
Overall this is an excellent and interesting paper. Please consider the use of "poor" and "rich" as potentially value laden language. Given the quantitative nature of the work, could you simply refer to the quintile?
Author Response
15 June 2022
Dear editor and reviewers, IJERPH
Many thanks for the comments and suggestions by the academic editor and the two reviewers; we found them very useful and constructive in shaping the revision of our manuscript. We had responded point by point to all comments in the revised manuscript. Please find our responses below.
We, therefore, submit three files
- Point-by-point responses to the editor and reviewers, which is this file
- Revised manuscript in track change
- Revised manuscript in clean text.
Best wishes,
Thitikorn Topothai
The corresponding author on behalf of all authors.

Reviewer 2 Report
Thank you for the opportunity to review this article. The authors have successfully identified four factors (ie., gender, family wealth, parental education, and parental interaction) that influenced the development of Thai children. The research aims were clear and supported by relevant literatures. Methodology and data analysis were comprehensive, and were documented to permit transparency of the research. The sample size was sufficient to justify the analysis and the conclusion. Several relevant discussions were held to support future research and policy review/implementation. The academic presentation and style were coherent and clear. Congratulations to this important study!
I recommend publication of this study with the following minor revisions:
p. 8 “greater chance to achieve an appropriate ECD index in children living with parents with higher education” & p. 11 "recommended to promote parental interactions through play with young children" -- provide specific & implementable recommendations to enhance accessibility & quality of education in Thailand and to provide contextually relevant guidelines for policy development to improve family wealth.
p. 8 “Approximately 92% of Thailand's children under the age of five had an appropriate ECD index…” – discuss implications/recommendations of such findings [e.g., to compare with previous data such as 93% in 2012 & 2016]
p. 9 “Further qualitative research, in addition to the regular quantitative surveys through MICS….” – please recommend specific qualitative design/approach for future study based on the context of Thailand
P. 8 “…A study in Thailand showed higher stunting among children living in the least well-off households” – further discuss the link/s between the cited study & the findings of the present study (ie., height for age & stunting in Table 2)
P.1 “Thai children under the age of five”; p. 3 “children aged 3 to 4 years”; & ‘p. 8 “children aged 24–59 months” – check consistency of the age group of the present study
Author Response

(The authors gave the same response as above.)
